# The Relations between Repetitive Behaviors and Family Accommodation among Children with Autism: A Mixed-Methods Study

**DOI:** 10.3390/children10040742

**Published:** 2023-04-19

**Authors:** Gal Shiloh, Eynat Gal, Ayelet David, Elkana Kohn, Ariela Hazan, Orit Stolar

**Affiliations:** 1Department of Occupational Therapy, Faculty of Social Welfare & Health Sciences, University of Haifa, Haifa 3490002, Israel; galweinman@gmail.com (G.S.);; 2Clinical Pharmacology and Toxicology, Shamir Medical Center (Assaf Harofeh), Zerifin 6093000, Israel; elkanak@shamir.gov.il (E.K.);; 3The Autism Center/ALUT, Shamir Medical Center (Assaf Harofeh), Zerifin 6093000, Israel

**Keywords:** children with autism, repetitive and restricted behaviors and interests, family accommodation

## Abstract

Restricted and repetitive behaviors and interests (RRBI) are a significant component in diagnosing autism spectrum disorder (ASD). They often pose the main challenge in day-to-day functions for children with ASD and their families. Research addressing family accommodation behaviors (FAB) in the ASD population is scarce, and associations with the characteristics of the children’s behaviors are unclear. This sequential mixed-methods study assessed the correlation between RRBI and FAB within the ASD group to deepen the understanding of parents’ subjective experiences regarding their children’s RRBI. It included a quantitative phase with a follow-up qualitative study. A total of 29 parents of children with autism (5–13 yr) completed the study questionnaires; a total of 15 also were interviewed regarding their children’s RRBI and related FAB. We used the Repetitive Behavior Scale-Revised (RBS-R) to assess RRBI, and the Family Accommodation Scale (FAS-RRB) to assess FAS. In-depth interviews from phenomenological methodology were used in the qualitative phase. We found significant positive correlations between the RRBI and FAB overall and their subscores. Qualitative research supports these findings, adding descriptive examples of the accommodations families make to address the RRBI-related challenges. The results indicate relations between RRBI and FAB and the importance of practically addressing children with autism’s RRBI and their parents’ experiences. Both affect and are affected by the children’s behaviors.

## 1. Introduction

Family-centered therapy places the family at the center of the therapeutic intervention; it is based on the assumption that the child develops best when the entire family’s needs are considered. Family-centered care is characterized by applying respectful attitudes toward families, sharing information that enables informed decisions, and addressing family choices and priorities by creating teamwork and an equal partnership with the entire family [1,2]. 

Autism spectrum disorder (ASD) is a lifelong neurodevelopmental disorder with unique characteristics manifested in various areas, including social-communication and behavioral problems [3]. According to the Centers for Disease Control and Prevention [4] data, 1 in 36 children in the United States is diagnosed with ASD. Two of the clinical diagnostic characteristics of ASD are deficiencies in (a) social communication and interaction and (b) restricted and repetitive behaviors and interests (RRBI) [3], (p. 50).

RRBI are behaviors related to repetition, rigidity, fixation, and resistance to change [3]. Not specific to ASD, RRBI are also found among infants and even adults with typical development or different developmental delays [5,6]. However, children with autism consistently express higher levels and wider RRBI ranges than others [7].

These behaviors are often grouped into two categories: (a) low-order RRBI, which include repetitive movements of body or objects and sensory-seeking behaviors, and (b) high-order RRBI, which include restricted interests and ritualized verbal and behavioral patterns. This classification usually refers to a child’s functional level in expressing a specific behavior. Low-order RRBI are usually present in younger children with developmental delays and lower cognitive abilities, and high-order RRBI in children with higher cognitive abilities and verbal skills [8,9].

RRBI have been described as affecting various daily functions, such as self-care, social occupations, and activities. For example, a child with autism may insist on wearing a favorite shirt every day, regardless of the season and weather [10,11]. The behaviors may affect the children’s and family members’ participation in common situations and require daily changes and modifications. For example, a child with ASD may feel compelled to take the same route from home to school, eat the same foods with the same utensils every meal, repetitively play with the water tap instead of brushing their teeth, or perform a ritual of shutting off the light when leaving a room regardless of whether other people are still there [11,12]. 

These behaviors can result in strong stigma and social rejection [10]. They might pose challenges to involvement in social occupations and limit the development and implementation of acceptable, compatible regulation means that would be more flexible or age-appropriate [13]. For example, the high focus on specific topics under restricted areas of interest might limit participation in situations that foster acquiring and developing social or learning skills. They, thus, reduce opportunities to expand learning [14]. 

The professional literature offers extensive reasons and theories regarding the etiology of RRBI. Some researchers believe that organic impairments such as structural defects or brain functions may be possible factors of RRBI [15]. In addition, RRBI may manifest differently depending on specific characteristics such as age, gender, and cognitive functioning [16]. Society often perceives repetitive behaviors as inappropriate or without purpose. However, studies and interviews with people with autism show that RRBI behaviors play a role in promoting participation and attention in situations of sensory overload and high stress [10]. Furthermore, when restricted interests are viewed as a strengthening and motivational factor, they can provide options for developing conversational skills around the topics of special interests, a deeper understanding of topic-related situations, and even career opportunities in the specific field [5,10,13]. Therefore, therapists working with children with autism must understand what these behaviors mean to the person who performs them. This requires examining RRBI in their specific context and determining whether they allow for or limit participation in daily life [13]. It is also important to identify and address challenges that may result in the need for families of children with autism to develop accommodations.

Family accommodation behaviors (*FAB*) are strategies designed to facilitate the child in dealing with challenging situations [17]. They are defined as functional responses or adjustments of the family to the daily life demands of the child with the developmental disability [18,19]. Two definitions for FAB can be found in the literature. The first, concerning the literature on children with developmental disabilities, notes *FAB* as a positive means of enabling the family’s well-being [18]. Families act or behave in response to factors in their social and physical environments over which they have only partial control. Such actions aim toward constructing a stable sociocultural environment related to routine, meaningful family occupations. Parents may construct their family’s goals, habits, and routines with the aim of enabling the child’s participation in his environment. These include services provided to the family, residence, living standard, childcare tasks, parental roles at work and home, and mental and physical support for the family. Studies on parenting children with disabilities have indirectly noted how FAB increased adjustments for childcare in daily routines, e.g., [18]. The second definition of FAB, found in studies related to obsessive compulsive and anxiety disorders, refers to family members’ behaviors aiming to prevent the child from responding with distress or anger outbursts to actual or anticipated exposure to a stressor [19,20]. Common examples include changing family plans to accommodate the child and providing reassurance and active involvement in rituals or repetitive behaviors [21]. Although FAB are protective by nature, they may often be overprotective. They may result in reduced opportunities for learning and development in the context of emotional regulation when facing the stressor or the development of adaptive coping strategies with anxiety-causing situations [22]. The two FAB definitions can be a conflict for therapists treating families of children with disabilities. According to the first definition, FAB might help the therapeutic intervention and guidance for parents and promote the child’s functioning; however, according to the second definition, FAB may be obstacles that can prevent the child’s participation in their environment.

The symptoms and comorbidities associated with autism can cause complex parenting demands [23]. Recently, studies began to address FAB among children and youth with autism. Storch et al. [20] found that 97.5% of parents of youth with ASD and anxiety disorders reported certain FAB at least weekly. Accommodation behaviors were associated with higher symptom severity and functional impairments [24]. Parents of children with autism might adjust activities’ demands, reduce exposure to stressors, and avoid unexpected situations that could provoke the children’s problematic behaviors [17]. Therapists who work with families of children with autism must understand and address these behaviors because they significantly affect the child and therapeutic intervention.

Although understanding the relationships between FAB and specific RRBI could allow for interventions tailored for children with autism and their families, scholars have not yet examined such relationships. Therefore, this study aims to examine the relations between RRBI and FAB in children with autism. The qualitative part of the research attempts to answer these questions:How do parents of children with autism perceive and experience their children’s participation in everyday life?How do parents of children with autism describe the family accommodations they make in daily life?

## 2. Methods

### 2.1. Participants

We used a sequential mixed-methods explanatory design model [8,25], collecting data at two time points: The quantitative phase (January 2019–February 2020) and then the qualitative phase (March–August 2021). Through this model, it was possible to obtain a structure of quantitative data related to the children with autism’s frequency and nature of RRBI and the frequency and types of FAB that were a part of families’ lives. In-depth interviews reinforced these data. The interview-guide questions were developed based on the quantitative results, giving them additional validity and expansion [26].

### 2.2. Procedure

Quantitative Phase. The quantitative dataset included the parents of 29 children with autism selected by nonprobability convenience sampling. Inclusion criteria were (a) parents or caregivers (herein of children aged 5 to 13 yr with (b) medical diagnoses of ASD accepted by the Ministry of Health, confirmed using the Autism Diagnostic Observational Schedule) [27]. Parents/caregivers of children with known genetic syndromes that cause autistic symptoms or diagnosed metabolic diseases and parents/caregivers with severe mental health problems (e.g., psychosis, drug addiction) were excluded.

Qualitative Phase. We invited parents/caregivers who participated in the first (quantitative) study phase into a qualitative follow-up phase. The first-phase researchers met the participants several times and created a list of parents who speak Hebrew as their first language. This list formed the basis for telephone contact with potential participants for the interviews. The first 15 consenting volunteers (no caregivers, but parents of 14 boys and 1 girl) participated.

We stopped interviewing after the 15th participant due to content saturation—the emerging data added no new themes [28]. We conducted deliberate sampling according to quantitative phase criteria, adding the following criteria: (a) We suggested that the caregiver or parent who was most actively involved in the child’s daily life participate in the interviews. (b) The parents’ Hebrew-language skills were sufficient for an in-depth interview. The qualitative study used a phenomenological approach [29] to examine the parents/caregivers’ experiences and perceptions of their children’s RRBI and FAB. We conducted and audio-recorded the interviews using secure Zoom Video Communications software (Premium version) +. Each interview lasted 35 to 60 min (M = 46 min).

### 2.3. Measures

Medical Demographic Questionnaire. This questionnaire collected demographic and medical information, including gender, age, complex medical conditions, and medications.

Repetitive Behavior Scale-Revised. The Repetitive Behavior Scale-Revised (RBS-R) [30] examines the frequency and severity of recurrent behaviors in the previous month based on parent reports. It includes 43 items rated on a 4-point Likert scale from 0 to 3. The items form six subgroups: Stereotyped behavior, Self-injurious behavior, Compulsive behavior, Ritualistic behavior, Sameness behavior, and Restricted behavior. The final score, the sum of all items, ranges from 0 to 129. Higher scores indicate higher RRBI frequency. 

The questionnaire was found to have internal consistency (α = 0.78–0.91) [31] and test–retest reliability (α = 0.52–0.96) [32]. We found high total-score (α = 0.929) and subscore (α = 0.911–0.633) internal reliability in this study.

Family Accommodation Scale for Restricted and Repetitive Behaviors. The Family Accommodation Scale for Restricted and Repetitive Behaviors (FAS-RRB) [19] is an adapted version of the family accommodation scale used in FAB studies in obsessive compulsive and anxiety disorders [24]. The questionnaire assesses FAB due to RRBI in children with autism. Respondents rate items on a 5-point Likert scale from 0 (never) to 4 (daily). The first 7 items, related to FAB frequency, are summed to a general accommodation score from 0 to 29. The eighth item addresses parental distress the FAB might cause, and the last three relate to the child’s short-term response if parents make no accommodations. 

The first seven questions were found to have high internal reliability (α = 0.935) [19], as in our study (α = 0.809). The last three questions also were found to have high internal reliability in our study (α = 0.768).

Interviews. The in-depth, semi-structured interviews were based on a guide compiled for the qualitative part of this study. The guide uniformly framed all interviews with mostly open-ended questions on the study’s topics. The questions focused on RRBI behaviors (e.g., “Some parents report difficulties with rigidity in transitions between activities, environments, and various rituals. Can you share examples of similar behaviors?”), different functional areas (e.g., “Can you describe your child’s sleeping habits?”), and FAS in their home (e.g., “Can you describe the behaviors’ influence on you and the other family members? On your and other family members’ accommodations? On the home routine?”).

The interviewer was a PhD student and occupational therapist with experience in treating and guiding parents of children with autism. The interviewer telephoned parents who had responded to her invitation to participate and provided full information about the study’s purpose and steps, confidentiality, and contact methods.

### 2.4. Data Analysis

We analyzed the quantitative and qualitative data at different times. We first processed the quantitative data using SPSS (ver. 27) and the demographic characteristics using descriptive statistics. Due to the small sample size and abnormal study variable distribution according to the Shapiro–Wilk test, we used Spearman tests to correlate asymmetric variables, examining relations between RRBI total and sub-scores (per RBS-R) and FAB (per FAS-RRB).

After completing and transcribing the interviews, we used a systematic phenomenological approach through three steps [29], creating: (1) thematic codes by identifying significant text units, (2) seven categories that include similar or different characteristics by creating groups and mapping the thematic codes, and (3) three conceptualized major themes by unifying the categories [33]. Three authors coded the first three interviews independently to create initial categories and map the other interviews.

To help detect possible bias and help achieve data trustworthiness, we presented in-depth quotes from the original transcripts (using encrypted location), journaled our thoughts and feelings during the interviews, and documented detailed descriptions from the process [33]. We conducted lengthy open discussions and data peer reviews, supported by our familiarity with the professional literature and themes discussed in the interviews, until we reached agreement.

## 3. Results

### 3.1. Quantitative

All participants in the study were the children’s mothers—28 were married (1 single mother)—and Jewish; a total of 52% had a non-religious lifestyle, 24% had a traditional lifestyle, and 17% were ultra-Orthodox (7% did not report). About 43% of the mothers had two children, 29% had three, and 25% had more than three (for 3% of the mothers, the child with autism was the only child). The 29 children with autism (*M* = 7.7 years, *SD* = 0.41) were primarily male (*n* = 24), representing the high prevalence of males within the ASD population. About 77% of the children demonstrated a high level of ASD symptom severity, 19% demonstrated moderate severity, and 4% demonstrated low symptom severity. Table 1 presents descriptive data with the RBS-R overall and sub-scores. Ritualistic behavior (*M* = 1.29, *SD* = 0.17) was the most prevalent, and self-injurious behavior (*M* = 0.44, *SD* = 1) was the least prevalent RRBI.

Per the FAS-RRB, parents most often reported (84% generally, 48% daily) using the FAB of providing special objects to address their children’s RRBI; a total of 91.6% (33.3% daily) actively assisted their children in avoiding a stimulus, resulting in RRBI. Many parents reported changing their schedules to accommodate their child’s behavior: 52% reported changing work habits (13% daily), and 86.95% changed leisure habits (26.1% daily). Parents also reported avoiding people and places and frequently changing the family’s schedules. Table 2 shows that most (82.6%) reported parental distress following accommodation but that, without accommodation, their children might experience distress (82.6%), respond aggressively (73%), or increase RRBI (75%).

Considering the small sample size, we performed a Bonferroni correction for the correlations that emerged to increase confidence in the results. The correction was performed by dividing the significance level (0.5) by the number of correlations (18). Thus, only correlations with significance levels less than or equal to 0.003 are marked in Table 3. Those correlations revealed that FAS-RRB parental distress was significantly correlated with FAB frequency (*r* = 0.620, *p* = 0.002) and the child’s negative response without accommodation (*r* = 0.642, *p* = 0.001). We noted a significant positive correlation between the overall RBS-R score and FAS-RRB accommodation score (*r* = 0.607, *p* = 0.001)—the more RRBI the children demonstrated, the more FAB their families exhibited. Table 3 shows that the FAB score significantly correlated with Compulsive behavior (*r* = 0.649, *p* < 0.001), Ritualistic behavior (*r* = 0.579, *p* = 0.002), Sameness behavior (*r* = 0.578, *p* = 0.002), and Restricted behavior (*r* = 0.608, *p* = 0.001). Families of children who demonstrated these RRBI behaviors performed FAB more frequently.

### 3.2. Qualitative

The results addressed challenges the children with autism and their families experienced from the parents’ viewpoints. These challenges were reflected in the children’s ability to participate and the FAB the family made. Three themes emerged from the interviews (Table 4): (a) child’s challenges, (b) parent’s and family’s challenges, and (c) “You just can’t arrange the child’s path for him all day”.

#### 3.2.1. Child’s Challenges

All participants described challenges related to the children’s RRBI. Among the descriptions were behaviors that can be attributed to low-order RRBI (e.g., engaging in sensory seeking and repeated movement of body or objects) and to high-order RRBI (e.g., need for sameness and rituals, and intensely, unusually engaging in areas of interest). For example, R, the mother of a 6-year-old, interpreted her son’s repetitive behavior as a calming source of pleasure from repeated auditory stimulation: “He likes to open and close doors. He likes the noise of slamming doors. He … repeats it over and over again”.

Participants also shared their children’s struggles with daily activities due to their need for sameness and rituals, to go places only with a particular object, or to dress, eat, or shower only in specific ways. S described her 9-year-old’s ceremonial morning behavior: “He was not willing to drink chocolate milk if he had socks on … or all sorts of schemes like that”. 

The challenging behaviors related to intense engagement in areas of interest. N described her 11-year-old’s interest in collecting CDs as so compulsive that it was uncontrollable: “He really likes CDs. We have a pile of several hundred CDs in the house. Basically, every time we approach a store that has CDs … he snatches the CD and runs away with it. In this example, N illustrated the intense interest leading to safety issues. Such behaviors present severe challenges for the children and family members. 

#### 3.2.2. Parent’s and Family’s Challenges

All participants described difficulty coping with their children’s nonadaptive RRBI. Throughout the interviews, they shared feelings of guilt, marital and family stress, and ongoing need for self-regulation, given their difficulties in parenting a child with autism. Most negative feelings resulted from their need to perform FAB continuously throughout the day. S described her 9-year-old’s need for one parent’s presence when lying down to sleep: 

We would just sit next to him for long hours. … It was a nightmare. … One of us sits until 11:30 p.m., looks at the ceiling, and wants to die. … Many times, we say it’s a miracle we did not get divorced.

Family events with relatives can be complex and require different preparation. S shared the struggles of needing to prepare her 9-year-old for adapted behavior before attending social events:

There was much alertness about what he would do. We would also literally tell him before [events], “You can’t touch anyone; you can’t sit down [on anyone].” And that created a lot of tension, yes. Lots of adjustments. A lot of such preventive conversations. That’s terrible. To tell your child what he should not do all day.

Involvement in adaptations for a child with autism may increase arguments, fights, and tension within the home, including the entire family. S described the tension among the family members due to their accommodation behaviors:

I had to come with him everywhere. To the bathroom. … We have a fridge on the balcony so he wouldn’t go out to the fridge, which is like … four meters, like, it would create a lot of tension inside the house because I can’t always come with him to the bathroom, and then I would ask his sisters, and then they would get angry, and it created a kind of something so terribly complex.

#### 3.2.3. “You Just Can’t Arrange the Child’s Path for Him All Day”

The challenges that the children’s behaviors posed affected the whole family. Efforts to understand a child with autism’s needs while meeting other family members’ needs led the participants to search constantly for ways to accommodate them. The varied FAB they presented in the interviews required the parents’ adaptation to the children’s needs. L described accommodations to her 7-year-old’s clothing to deal with his propensity for masturbation: “Such masturbation he performs all the time. … We had to sew his clothes, his shirt, sew it to his pants and add a zipper at the back because it was always there”.

Other FAB examples related to household members’ daily routines. R described difficulty producing a “surprise-free routine” for her 5-year-old: 

Every little thing can infuriate him. If the phone suddenly does not work, there is no game he could download, the neighbor turned on the elevator light before him … if it’s a light in the elevator or if it’s everything, … a computer that does not work, if I promised him something and, in the end, it was canceled. … You just can’t arrange the child’s path for him all day so that everything will go smoothly.

G described the difficulty in dealing with her 12-year-old daughter’s rigidity and ritual:

She needs preparation in advance. She needs preparation for everything. You cannot spontaneously say now, “Let’s go.” It doesn’t work. She needs to be guided and what is going to happen and what is the schedule. Changes. She does not accept them properly, so to speak.

Two-thirds (*n* = 10) of the parents reported difficulties leaving the house as a family, posing the dilemma of bringing the child with autism or giving up events altogether. M, the mother of an 8-year-old child with autism, explained the accommodation behaviors her family was forced to perform by either giving up time together or coordinating between the spouses to divide out-of-home supervision:

We tried to adjust ourselves to it. If there are places not suitable to go with him, then it is not suitable to go with him. For example, crowded places, places with a lot of people, with a lot of noise. So we tried not to go or go with him only if one [parent] would guard him. So yes, it changes our lives a bit. Yes, we need to adjust ourselves.

M later described adjusting family vacations to meet her 8-year-old’s challenges:

If he’s making noises because he’s turning around, because he’s shouting, he’s—whatever, it’s something interfering. … So, on family vacations, we prefer instead of taking a hotel to stay at a guest cabin, something alone. We don’t mix with other families!!!

The participants’ statements in this section may indicate the broad implications a child’s behavior has for creating and preserving accommodation behaviors and the implications of FAB on the family’s participation with relatives or community events. Hence, this section suggests a mutual relationship between FAB and child affecting each other. 

The parents often used their self-regulation techniques to cope with their children’s behavioral difficulties. Some shared strategies of patience and compassion. M described her need for self-regulation to deal with her 10-year-old’s challenging behaviors:

Today [the child] did not want to brush his teeth because he has a loose tooth. … I’ve had enough, and I don’t have the strength to say to him one more time, “Well then, brush here, brush here.” … I told him that dozens of times already. … Then it comes out aggressively, and it’s unpleasant, and it’s not fun. My accommodations are to constantly try to calm myself, to be the calm person in our home that always goes, “Come on, let’s do that”. 

The parents also mentioned professional support and its effect on their FAB and the children’s behaviors. M shared coping with her 7-year-old child’s behavioral problems and how professional guidance helped her make a difference:

There is also a lot of behavioral, emotional, work that means a lot. … We met a therapist by chance, and he came to us. … He just gave us tools. … We saw things a different way, … other tools that showed us how to respond. Then there was the lockdown [COVID] and an opportunity for my husband to really do a so-called focused intervention. It made a big difference. After the lockdown, [the child] went back to school a different child. Previously, the teacher pressured us to send him to a psychiatrist, but, thanks to our attitude change, we could deal with it.

## 4. Discussion

This study aimed to examine the interrelationships between RRBI and FAB among children with autism and their families. The results support the significant impact of RRBI on children’s functioning and their families’ daily lives.

In this study, all parents reported using FAB at least monthly (76% daily). In Feldman et al.’s [19] study investigating FAB among parents of children with autism in the context of RRBI, 80% of the parents reported using FAB at least monthly (55% daily). Those results were similar but slightly lower than our study’s results. The findings from the interviews supported this result and detailed the various FAS that occur daily and continuously at home.

This study found a strong, positive, and significant relationship between RRBI frequency (according to the RBS-R overall score) and FAB occurrence (according to the FAS-RRB overall score). We specifically found strong correlations between higher-order RRBI, such as compulsive, ritualistic, sameness, and restricted behaviors. Higher-order RRBI are attributed to children with autism with higher cognitive and verbal abilities [9,34]. Their environment, therefore, might expect them to engage in more adapted behaviors. Hence, parents may invest efforts into FAB to assist their children’s functioning within the family and community. In the interviews, the participants greatly elaborated on FAB related to their children’s various higher-order RRBI behaviors. For instance, they tried to create routines as regular as possible due to the child’s difficulty with changing daily routines or they adjusted the clothes of a child who experiences compulsive masturbation).

A different explanation could be the family’s tendency to perform FAB. Possibly within families that demonstrate FAB at a high level, children develop many RRBI behaviors with a higher frequency. Parental behavior greatly influences the way psychopathologies are presented and expressed in childhood, and different parental behaviors may have an impact on the development of different behavioral patterns, including RRBI [35,36].

In contrast, our study revealed no correlations between low-order RRBI and FAB, although children in this study presented such behaviors. These results suggest that parents were more prone to accommodate higher-order than lower-order RRBI. Past studies suggested that the children’s levels of impairment and environmental demands might affect the FAB nature and frequency [18]. 

The parents’ perceptions of their children’s behaviors might also play a role in their decisions to react with FAB. For example, past research suggested that children’s RRBI play an essential role in regulating and lowering their stress and anxiety levels [12,14]. Parents might perceive lower-order RRBI to have a functional effect because their calming functions assist in self-regulation, even if the behaviors seem nonadaptive to environmental demands. In the interviews, the participants interpreted their children’s repetitive behaviors as regulating and calming (e.g., repeatedly closing and opening doors). Further, intervening to reduce these behaviors is complex and often unsuccessful [13]; hence, parents avoid trying to change them.

Past studies showed that parents invest numerous efforts in FAB [17,20]. Results from our study’s quantitative part and interviews in the qualitative part support those findings. The parents’ efforts include adapting the physical environment (e.g., changing the nature of a family vacation to a remote cabin instead of a crowded hotel) and limiting exposure to stressors to avoid frustrating the child or family (e.g., foregoing family outings due to the child’s difficulty with noises). Past research also described FAB as immediately easing problematic situations and contributing to family members’ mental well-being and participation in daily functions [37]. One mother in our study described how she intensively prepared and explained in detail every future event to allow for her daughter to participate in the family’s daily life.

Although FAB may provide immediate relief for the child and family, they sometimes present challenges: The parents’ efforts may result in parental distress [38,39]. Our quantitative results showed that 82.6% of parents expressed “parental distress following accommodation” for their child’s RRBI. Participants shared their negative feelings about FAB that reduced family time and about constantly adapting to their child with autism’s challenging needs. The literature suggested that families pay an emotional price following FAB, such as high anxiety and signs of depression [17].

FAB also considerably burden families in the long run because they require effort to reorganize family activities around the child’s needs [39]. In this study, parents most often reported avoiding bringing children with autism to family gatherings or accustoming them to significant support when performing daily functions. These, in turn, could affect the child’s participation in diverse life areas. The parents and siblings eventually pay a considerable price. These FAB also may negatively affect the child with autism because they could reduce opportunities to develop emotional regulation or adaptive coping strategies in anxiety-causing situations [22].

In addition to the child’s characteristics and behaviors, studies have examined the parent’s variables that may predict a high frequency of FAB, especially among families of children with anxiety disorders [40,41]. For example, parents with psychopathologies such as anxiety disorders are more likely to perform FAB to a higher degree than parents without psychopathologies [42,43]. However, even among parents without diagnosed disorders, different parenting approaches [36], forward thinking about behavioral consequences [40], ability to regulate emotions [44,45], high parental empathy [40,46], and believing that anxiety is harmful to the child [41,45] all have a great impact on the possibility of these parents’ FABs. In some cases, the presence of a particular parental trait can increase or decrease the FAB frequency. For example, several studies described high parental empathy as a factor increasing the likelihood of having FAB to a high degree because the parent may identify with the child’s feelings [40]. However, Settipani and Kendallit’s study [46] found that high empathy is actually related to greater reactive flexibility and fewer FAB behaviors in low-stress situations. They hypothesized that the parents know how to allow for their child to deal with situations they are capable of handling and promote their child’s functioning and participation in a gradual manner. Other courses of action might benefit the children and their families in dealing with these challenging behaviors. Parents in our study described how proper guidance from a therapist and implementing interventional tools at home significantly reduced their children’s behavioral problems at school. This example suggests that focusing on changing parental responses in the home environment might contribute to changes in the children’s behavior, as other studies supported [35,47].

The research literature presents various methods and approaches to the treatment and reduction in FAB among families of children with anxiety disorders or OCD. The main one is cognitive behavioral therapy, which focuses on strengthening the child’s and parent’s ability to deal with stressful events [20,48]. In addition, there are different types of parent training programs and behavioral approaches to integrating learning [40,41,49].

Limitations and Future Research. Several limitations of the current study should be addressed. The sample was small and relatively homogeneous. Because we recommended the participation of the one parent who was more involved in the child’s daily life, all interviewees were mothers. Family accommodations are related to norms and habits that the wider society may also influence. Thus, a larger, diverse sample (e.g., various marital statuses, cultures, and fathers and caregivers) might improve our ability to investigate interrelationships between RRBI and FAB and deeply understand the FAB issue. In addition, characteristics that might affect the relationship between FAB and RRBI (e.g., ASD severity or the children’s or parents’ anxiety characteristics) were not measured or taken into account in this study. Further research with a wider, more heterogeneous sample is needed to better understand these interventions’ implications for the children’s and parents’ well-being and effectiveness in the short and long terms. 

## 5. Conclusions

This study’s findings revealed associations between RRBI and FAB. In-depth interviews further explained this relationship, shedding light on the effects of FAB on children and their families. The relationship may suggest that FAB not only respond to but also affect the children’s behaviors and the family’s quality of life.

The results may have clinical implications for therapists. Given the correlations between FAB and RRBI and the hypothesis that there is a bidirectional effect of the parents’ FAB on the children’s RRBI, therapists who guide parents to adjust to encourage their child with autism’s participation should understand the differences between adjustment and FAB. This study also suggests therapists perform therapeutic interventions focused on the parents and their characteristics to reduce FAB and possibly reduce behaviors related to RRBI. Therefore, a comprehensive assessment of the parents’ characteristics, such as the presence of parental anxiety patterns, knowledge, and beliefs about the nature of RRBI of the children, degree of parental empathy, and current abilities and ways of coping, may assist in the development and accuracy of specific family-focused interventions. Among other things, they can raise awareness and educate parents about the possible consequences for FAB in the long term and help families discover alternative methods to promote coping with the challenging everyday behaviors of their child with autism, thus benefitting the children and their families.

## Figures and Tables

**Table 1 children-10-00742-t001:** Descriptive Statistics: RBS-R and FAS-RRB Questionnaires.

	Variable	*N*	Range	*M*	*Mdn*	*SD*
RBS-R	Total score	29	0.09–2.15	0.88	0.78	0.1
	Stereotyped behavior	29	0.00–2.33	0.79	0.67	0.12
	Self-injurious behavior	29	0.00–2.63	0.44	0.29	0.1
	Compulsive behavior	29	0.00–1.88	0.61	0.5	0.1
	Ritualistic behavior	29	0.00–3.00	1.29	1.33	0.17
	Sameness behavior	29	0.90–2.55	1.13	1.18	0.14
	Restricted behavior	29	0.00–3.00	1.18	1	0.15
FAS-RRB	Accommodation score	26	0.43–4.00	2.11	2	0.21
	Parental distress	23	0.00–4.00	2	2	0.31
	Child’s short-term response	26	0.00–4.00	2.1	2	0.25

Note. RBS-R = Repetitive Behavior Scale-Revised; FAS-RRB = Family Accommodation Scale for Restricted and Repetitive Behaviors.

**Table 2 children-10-00742-t002:** Frequencies and Accommodation Percentages: FAS-RRB.

Accommodation	Frequencies (%)				
	Never	1–3 Times/Month	1–2 Times/Week	3–6 Times/Week	Daily
RRBI-related object	16	16	4	16	48
RRBI-related action	29.2	12.5	8.3	25	25
Avoiding RRBI-related stimuli	8.33	20.84	25	12.5	33.33
Avoiding activities due to RRBI	25	16.67	12.5	12.5	33.33
Changing family schedule due to RRBI	28	20	12	12	28
Changing work habits due to RRBI	48.8	17.4	17.4	4.4	12
Changing leisure habits due to RRBI	13	30.4	17.4	13	26.2
Total (mean)	24.04	19.12	13.8	13.64	29.4
Parental distress due to FAS-RRB	17.4	26.1	21.7	8.7	26.1
Child distress	17.4	17.4	21.7	21.7	21.7
Child aggression	26.9	15.4	19.2	15.4	23.1
Aggravation of RRBI	25	12.5	12.5	16.67	33.33

Note. RRBI = Restricted and repetitive behaviors and interests; FAS-RRB = Family Accommodation Scale for Restricted and Repetitive Behaviors.

**Table 3 children-10-00742-t003:** Spearman Correlation Coefficients Between RRBI and Family Accommodation in Children with Autism.

RRBI Behavior	FAS-RRB Score		
	Accommodation	Parental Distress	Child’s Response
RBS-R total score	0.607 ***	0.248	0.315
Stereotyped behavior	0.355	0.009	0.071
Self-injurious behavior	0.248	−0.107	0.080
Compulsive behavior	0.649 ***	0.273	0.388
Ritualistic behavior	0.579 **	0.348	0.445
Sameness behavior	0.578 **	0.370	0.395
Restricted behavior	0.608 ***	0.311	0.435

Note. RRBI = Restricted and repetitive behaviors and interests; FAS-RRB = Family Accommodation Scale for Restricted and Repetitive Behaviors; RBS-R = Repetitive Behavior Scale-Revised. ** *p* ≤ 0.01, *** *p* ≤ 0.001.

**Table 4 children-10-00742-t004:** Themes and Subthemes from the Interview Analyses.

Theme	Subtheme	Number Times Theme/Subtheme Occurred	Example Quote
Child’s challenges	Low-order RRBI	5	He would bang his head against the wall, he would pull down his pants, he would cry and scream for hours.
	Daily activities	7	Dealing with the fact that he is not ready to wear clothes. He calls it “old with old and new with new.” He is not ready to wear an outfit that is not new from the last season.
	Areas of interest	2	He would sit and read books until the middle of the night. We would just go to bookstores and buy books, and that’s what he would do.
Parent/family’s challenges	Guilt	6	When you have an undisciplined child, you are angry at yourself, the environment is angry at you, you are like, … I don’t know, you are helpless.
	Stress	11	He’s stressing you out so much you can’t breathe.
“You just can’t arrange the child’s path for him all day.”	Family events	6	He no longer comes with me to events or things like that. I call a babysitter, or his older brother looks after him. I can’t take him.
	Environment modification	5	I brought a designer, and we made a bean-bag corner, which would be some kind of relaxation corner.
	Daily routine	7	We actually need to brush his teeth and count slowly to 30.
	Leaving the house	10	What was very difficult with him was going out, shopping, or going to the supermarket, or anywhere.
	Self-regulation	4	Wow, the level of patience required is endless.

Note. RRBI = Restricted and repetitive behaviors and interests.

## Data Availability

The data presented in this study are not publicly available due to ethical approval guidelines. The data can be requested from the corresponding author.

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
