# Peer review of "The Relations between Repetitive Behaviors and Family Accommodation among Children with Autism: A Mixed-Methods Study"

_children, 2023, doi:10.3390/children10040742_

Round 1

Reviewer 1 Report

The paper is interesting and the topic is in line with  the Journal scope. Authors explore the association between   RRB in Autism spectrum disorders and Family reactions (Family accommodation) with a quantitative-qualitative method.

The manuscript is  well organized and writing is clear, particularly in introduction when authors present the topic.  The cited literature and  references are adequate. The reviewer suggests  some minor additions  and/or clarifications in some point of the text.

Introduction

[lines 53-59] low-order and high-order RRBI are differentiated, but it is not clear whether and how this classification is used for the results. See, for example, line 254

[lines 87 and following] please insert a more restricted (operationalized) definition of FAB. The reviewer thinks that the cited factors are very wide, from cultural (macrosystemic)  to phsysical (prossimal) factors. It is suggested to replace “cultural” with “social” (i.e. communicative and/or relational factors).

[line 103] Accomodation behaviors: Add an operational definition with some examples.

[line 109]. Please change this statement “The causes and manifestations… are still unclear” since it does not correspond to what is known about the possible functions and forms of RRBI in ASD. See, for example, studies by Tian, Gao and Yang (2022) and Ulijarevic et al. (2022), just to name a few recent ones.

Method

Please correct the title paragraph 2. Materials and Methods with Methods

2.1 Participants and Procedure

Please separate the Participants from the Procedure in two sub-paragraphs.

The number of participants is  small (only mothers), but it is limitation is justified by the clinical characteristics of ASD children. In “Participants” section, the reviewer suggests to add information about a) family characteristics (i.e. marital status, single parents, siblings, instruction) and b) ASD children characteristics: gender and ASD severity, since ADOS is cited as measure for ASD diagnosis  and inclusion criterion ([line 131]

Procedure

The study design is adequate and the description of the procedure is clear (only minor additions are suggested). The results and tables are well presented.

Minor revisions:

Repetitive Behavior Scale-Revised: please add Alphas.

2.3 Data Analysis

[lines 192 and following]  The authors say that only three interviews were independently coded to derive the categories [line 196]. Please, explain better how the interviews were coded (how many independent coders?). Secondly,  was a concordance index conclulated? If it is not possible to calculate it, please recall this methodological weakness in the "Limitations" section.

Results

[Line 254]. As previously mentioned, explain this difference between low- and high-order RRBI classification and how it is considered in the results.

Discussion

It is suggested to cite and discuss studies clarifying the influence of FAB and RRBs in a interactional perspective too (that is, not only the relationships between RRBS and FAB). For example, it would be useful to mention studies exploring parent’s variables impacting FAB, such as perceived control on child's problem behaviours or distress.

Limitations and Future Research

As authors talk about the associations between FAB and anxiety in autism (line 102), emphasize that a limitation of current study is not having measured and taken into account the ASD severity and other comorbidities impacting RRBI.

Similarly, another limitation is not having measured some influential characteristics of the parents (see, for example distress, marital conflict, or stress).

Since the sample is small (only mothers), this limitation needs to be filled in future research

Conclusions.

What are the study implications for intervention? The reviewer suggests to expand this section, citing studies that highlight how parents can learn effective techniques for the management of RRBS (i.e. parent education, see for example, Harrop, Mc Bee and Boyd, 2006), and the effects of these interventions both on ASD problematic behaviours and caregiver’s functioning (e.g. reduction of stress, family conflicts and increase in parent’s perceived efficacy and coping)

Reviewer 2 Report

This is an interesting paper that addresses a significant issue in families’ experience of raising a child with ASD. The introduction begins with a description of the role of Occupational Therapists, which seems slightly out of place here. Either the authors might consider the applications of a family centered approach for any professional working with children with ASD or they should expand on why they are focusing specifically on OT.  The introduction is largely clear and well organized, with the exception of the last paragraph on page 2, which provides a rather confusing description of family accommodations. This is clarified partially in the opening paragraph of page 3, but the first paragraph should be clarified.  The CDC has recently updated the incidence of autism in children in the US to 1/34. This should be updated in the introduction.

The methods and results are clearly described and the tables are helpful. The quantitative portion is especially clear.  The qualitative portion of the study adds valuable context to the quantitative data but could be shortened depending upon the aims of the authors.  If the goal is to report on the mixed methods study described here, the qualitative section might be shortened and the paper submitted as a brief report. Alternatively, if the authors choose to expand their discussion to include ideas about how families might mange accommodations more adaptively, then the qualitative data is especially valuable. 

The discussion section could be elaborated further to provide more in-depth consideration of the results. The authors note that the lack of correlation between lower order RRBs and FABs may reflect the fact that lower-order RRBs are less interfering and that families view them as serving an adaptive function, often a self-regulating function. The significant correlation between higher order RRBs and FABs is more complicated to understand. It may be that in families who provide greater accommodation, children develop more significant RRBs. The authors note the reverse possibility, that when children present more significant and interfering RRBs, families are more likely to accommodate them. But either interpretation is plausible and might be considered. The fuller discussion of this possibility would also support a broader discussion of the need to help families respond differently to RRBs. Significant family accommodation does not appear to serve the interests of the child and is associated with more familial distress. The authors hint at the possibility of addressing family accommodation at the end of the paper. Discussion of that issue seems an important implication of this work and might be developed more.

The authors identify the major limitations of the study. As noted earlier, the paper could be published as a brief report, if some of the detail about the qualitative data was shortened. Alternatively, the authors might expand the implications of their work in the discussion section to support publication in its current form. In either case, the data is valuable and the topic is important.

Reviewer 3 Report

An at-times quite poignant mixed methods study of family accommodation to restricted and repetitive behavior in a small sample of mothers of autistic youth (N=29 with a subset of 15 qualitative interviews).   For those who work with the population of autistic children and their families, themes will resonate and interest level will be high.  The authors rightfully acknowledge the need for replication of these findings with a larger and more diverse sample (i.e., reflecting different marital statuses, cultures, fathers and other nonmaternal caregivers, etc.).  A more diverse sample might help them to better delineate their emerging typography of accommodation to higher vs. lower order RRB's and truly test their hypothesis about whether accommodation will have a stress-relieving or exacerbating effect on caregivers based on their attributions (e.g., accommodation will be less stress-inducing for mothers who view the RRB as a regulation strategy).  

I think the manuscript will be strengthened if the authors could opine in their Discussion on potentially helpful interventions (e.g., ABA - in particular helping families to resist intermittent reinforcement when their children tantrum; adapting Piacentini's work on family accommodation in OCD; habit substitution, etc.)  

Author Response

Response to Reviewer 3 Comments

Point: I think the manuscript will be strengthened if the authors could opine in their Discussion on potentially helpful interventions (e.g., ABA - in particular helping families to resist intermittent reinforcement when their children tantrum; adapting Piacentini's work on family accommodation in OCD; habit substitution, etc.)

Response: 

We have added information on existing FAB treatment and reduction methods (see lines 164-468)

The conclusions section was expanded and a more comprehensive reference to ways of intervention for therapists was added (see lines 486-499)